# CASC-AI: Consensus-aware Self-corrective Learning for Cell Segmentation with Noisy Labels

**Ruining Deng** [1,2]                                                    RUD4004@MED.CORNELL.EDU

**Yihe Yang** [1]                                                        YIY4007@MED.CORNELL.EDU

**David J. Pisapia** [1]                                                 DJP2002@MED.CORNELL.EDU

**Benjamin Liechty** [1]                                                 BEL9057@MED.CORNELL.EDU

**Junchao Zhu** [2]                                                      JUNCHAO.ZHU@VANDERBILT.EDU

**Juming Xiong** [2]                                                     JUMING.XIONG@VANDERBILT.EDU

**Junlin Guo** [2]                                                       JUNLIN.GUO@VANDERBILT.EDU

**Zhengyi Lu** [2]                                                       ZHENGYI.LU@VANDERBILT.EDU

**Jiacheng Wang** [2]                                                    JIACHENG.WANG.1@VANDERBILT.EDU

**Xing Yao** [2]                                                         XING.YAO@VANDERBILT.EDU

**Runxuan Yu** [2]                                                       RUNXUAN.YU@VANDERBILT.EDU

**Rendong Zhang** [2]                                                    RENDONG.ZHANG@VANDERBILT.EDU

**Gaurav Rudravaram** [2]                                        GAURAV.RUDRAVARAM@VANDERBILT.EDU

**Mengmeng Yin** [3]                                                     MENGMENG.YIN.1@VUMC.ORG

**Pinaki Sarder** [4]                                                    PINAKI.SARDER@UFL.EDU

**Haichun Yang** [3]                                                     HAICHUN.YANG@VUMC.ORG

**Yuankai Huo** [2,3]                                                    YUANKAI.HUO@VANDERBILT.EDU

**Mert R. Sabuncu** [1,5]                                                MSABUNCU@CORNELL.EDU

[1] *Weill Cornell Medicine, New York, NY 10021*

[2] *Vanderbilt University, Nashville, TN, USA 37215*

[3] *Vanderbilt University Medical Center, Nashville, TN, USA 37232*

[4] *University of Florida, Gainesville, FL, USA 32611*

[5] *Cornell Tech, New York, NY, USA 10044*

**Editors:** Accepted for publication at MIDL 2025

## Abstract

Multi-class cell segmentation in high-resolution gigapixel whole slide images (WSIs) is crucial for various clinical applications. However, training such models typically requires labor-intensive, pixel-wise annotations by domain experts. Recent efforts have democratized this process by involving lay annotators without medical expertise. However, conventional non-corrective approaches struggle to handle annotation noise adaptively because they lack mechanisms to mitigate false positives (FP) and false negatives (FN) at both the image-feature and pixel levels. In this paper, we propose a consensus-aware self-corrective learning that leverages the Consensus Matrix to guide its learning process. The Consensus Matrix defines regions where both the AI and annotators agree on cell and non-cell annotations, which are prioritized with stronger supervision. Conversely, areas of disagreement are adaptively weighted based on their feature similarity to high-confidence consensus regions, with more similar regions receiving greater attention. Additionally, contrastive learning is employed to separate features of noisy regions from those of reliable consensus regions by maximizing their dissimilarity. This paradigm enables the model to iteratively refine noisy

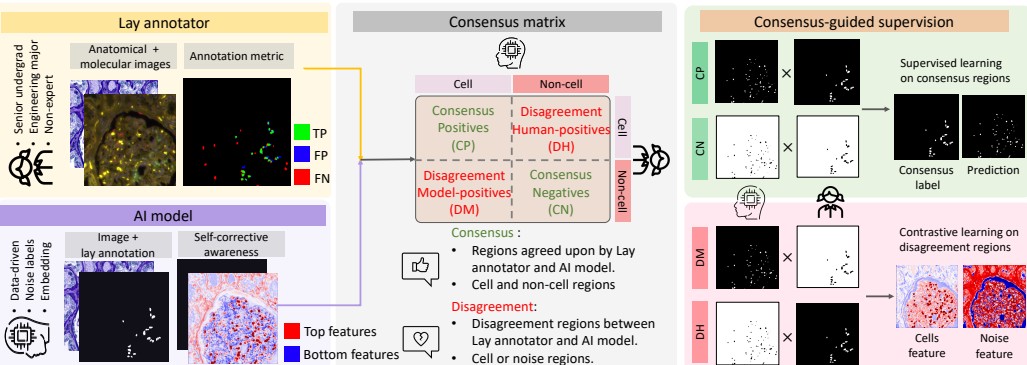

Figure 1: **Consensus-aware self-corrective learning.** We propose a Consensus-Aware Self-Corrective Learning for robust cell segmentation with noisy training data. The model leverages the CM to guide learning, prioritizing CP and CN regions with stronger supervision, while adaptively weighting DM and DH regions based on their similarity to reliable CP regions by contrastive learning.

labels, enhancing its robustness. Validated on one real-world lay-annotated cell dataset and two reasoning-guided simulated noisy datasets, our method demonstrates improved segmentation performance, effectively correcting FP and FN errors and showcasing its potential for training robust models on noisy datasets. The official implementation and cell annotations are publicly available at https://github.com/ddrrnn123/CASC-AI.

**Keywords:** Consensus matrix, Corrective Learning, Noisy label learning, Cell Segmentation

# 1. Introduction

Multi-class cell segmentation is essential for analyzing tissue samples in digital pathology, often serving as the initial step in extracting biological signals crucial for accurate disease diagnosis and treatment planning (Caicedo et al., 2017; Deng et al., 2020; Keren et al., 2018; Pratapa et al., 2021; Litjens et al., 2017; Border et al., 2024; Ke et al., 2023; Zhu et al., 2023, 2024). Accurate cell quantification aids pathologists in diagnosing diseases (Comaniciu and Meer, 2002; Xing and Yang, 2016), determining disease progression (Olindo et al., 2005), assessing severity (Wijeratne et al., 2018), and evaluating treatment efficacy (Jiménez-Heffernan et al., 2006). For instance, the distribution and density of cells in the glomerulus (e.g., podocytes, mesangial cells, endothelial cells, and epithelial cells) can serve as indicators of functional injury in renal pathology (Imig et al., 2022). However, cell-level characterization is challenging even for experienced pathologists due to the long annotation time, extensive labor required, significant variability in cell morphology (Zheng et al., 2021), and the potential for human error. Needless to mention the rigorous medical training required for a pathologist.

Previous efforts have democratized the annotation process by involving lay annotators without medical expertise and integrating pair-wise molecular images with pathological

images, resulting in a substantial number of accurate cell annotations for training AI models (Deng et al., 2023). However, this approach inevitably introduces noise and errors, necessitating correction by experienced pathologists. Directly training models on such noisy labels often leads to suboptimal performance. This highlights the urgent need for a corrective learning paradigm that effectively addresses label noise during cell segmentation model training (Vădineanu et al., 2022; Karimi et al., 2020). Previous research on noisy-label learning has focused on defining efficient loss functions (Zhang and Sabuncu, 2018; Wang et al., 2020; Ma et al., 2020) and leveraging multi-network strategies (Zhang et al., 2020b; Han et al., 2018; Lu et al., 2023; Guo et al., 2023). However, these approaches largely overlook the integration of feature-level analysis with pixel-level analysis to effectively identify annotation errors at the pixel level.

In this work, we propose Consensus-Aware Self-Corrective Learning (CASC-AI), which incorporates insights from the Consensus Matrix (CM) to guide its learning process (as shown in Fig. 1). Unlike conventional heuristic-based correction methods, CASC-AI actively learns from noisy annotations by leveraging both pixel-wise and feature-wise information to iteratively refine its predictions. The self-corrective learning mechanism autonomously detects patterns in annotation errors and adapts its training by distinguishing noisy labels from high-confidence regions through maximizing feature dissimilarity, thereby enhancing its robustness against annotation errors. The contributions of this paper is threefold:

(1) A Consensus-Aware Self-Corrective Learning is designed to provide robust cell segmentation when training data contains noise.

(2) A reasoning-guided noise-generation process is introduced for pathological cell images to simulate realistic noise for label analysis.

(3) By integrating Consensus Matrix insights at both the pixel and feature levels, the proposed method demonstrates improved segmentation performance, effectively addressing FP and FN errors, showcasing its potential for training robust models on noisy datasets.

## 2. Method

Introducing lay annotators into the labeling process significantly increases the volume of annotations available for training deep learning models. However, it also introduces noise and errors due to human visual limitations and variability among annotators. There are several types of annotation errors introduced by humans, including contour-wise boundary errors (Zhang et al., 2020a; Dang et al., 2024) and instance-wise location errors (Vădineanu et al., 2022; Goldsborough et al., 2024). In this study, we mainly focus on instance-wise location errors, where false positive and false negative cells are introduced during the molecular-empowered lay annotation process (shown in Fig. 1).

With the rapid development of deep learning, AI has demonstrated its capability in representing images (Oquab et al., 2023; Huang et al., 2021), providing reliable and stable latent features for image understanding. Therefore, the proposed CASC-AI aims to combine the strengths of human expertise and AI capability during the training phase, guiding the model to capture accurate information from lay annotations while distinguishing potential noise at the pixel level. The overall learning paradigm consists of three components: (1) Consensus Matrix, (2) Consensus-aware Supervision, and (3) Contrastive Noise Separation.

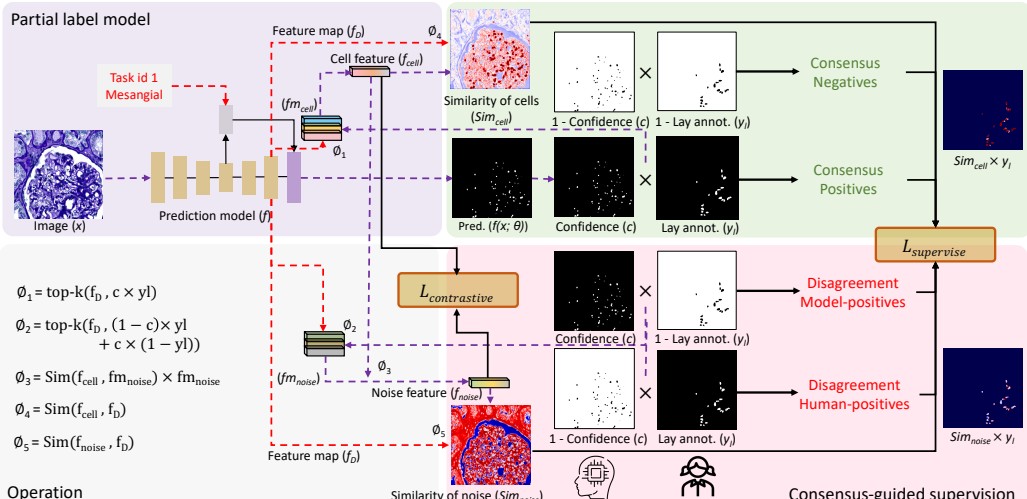

Figure 2: **Overview of the Consensus-Aware Supervision Framework.** The architecture integrates AI-derived confidence maps ($c$) and lay annotations ($y_l$) to identify consensus-positive (CP), consensus-negative (CN), and disagreement regions (DM, DH). This framework emphasizes robust training by focusing on regions of consensus and leveraging disagreement as informative cues for improved cell segmentation accuracy.

## 2.1. Consensus Matrix

To capture the agreement between lay annotators and the AI model, we define a Consensus Matrix (in Fig. 1), inspired by the confusion matrix, to guide pixel-level image understanding. The matrix is composed of the following components:

**Consensus Positives (CP):** Regions where both the AI and annotators agree on a "cell" annotation. These regions represent strong consensus for action, where both parties confidently identify cells.

**Consensus Negatives (CN):** Regions where both the AI and annotators agree on a "non-cell" annotation. These regions reflect mutual consensus to abstain from action, ensuring non-cell regions are left unannotated.

**Disagreement Model-positives (DM):** Regions where the AI identifies a "cell," but annotators label it as "non-cell." These regions highlight potential false negatives in the lay annotations, where cells may have been missed.

**Disagreement Human-positives (DH):** Regions where the AI labels a region as "non-cell," but annotators identify it as a "cell." These regions represent potential false positives in the lay annotations, where cells may have been overannotated.

## 2.2. Consensus-aware Supervision

Building on our previous works (Deng et al., 2023, 2024a), we select a token-based residual U-Net from (Deng et al., 2024b) as the backbone for cell segmentation tasks. This backbone demonstrates superior performance in multi-class cell segmentation using partially labeled datasets, compared to two other cell segmentation backbones (Hörst et al., 2024; Israel et al., 2024) as shown in Table 5. As illustrated in Fig. 2, the model outputs the final prediction logits $p \in \mathbb{R}^{2 \times W \times H}$, the pixel-level feature map of the decoder's last layer $f_D \in \mathbb{R}^{Ch \times W \times H}$, and a confidence map $c \in \mathbb{R}^{1 \times W \times H}$, which represents the foreground channel of $p$ after applying the channel-wise softmax function. $W$ and $H$ are the width and height of the input image, while $Ch$ represents the number of channels in the decoder's last layer. The confidence map $c \in (0, 1)$ indicates the confidence level of predictions: values closer to 1 suggest stronger confidence in identifying a region as a cell, while values closer to 0 suggest a higher likelihood of non-cell regions.

**Consensus Cell Feature Distillation:** Using the confidence map $c$ from the AI model, we combine it with lay annotations $y_l$ to identify pixel locations with the highest agreement scores $a_{\text{CP}}$ in CP regions. These regions are used to distill features $f_{\text{cell}}$ that best represent cell types. The computation for $a_{\text{CP}}$ and $f_{\text{cell}}$ is defined in Eq. 1 (annotated as $\phi_1$ in Fig. 2).

$$
\begin{aligned}
a_{\text{CP}} &= c \cdot y_l \\
\text{Ind}_{\text{CP}} &= \text{argsort}(-a_{\text{CP}})[: k] \\
f_{\text{cell}} &= \frac{1}{k} \sum_{i=1}^{k} f_D(\text{Ind}_{\text{CP}}[i])
\end{aligned}
\tag{1}
$$

**Disagreement Noise Feature Distillation:** In DH and DM regions, where the AI model and lay annotators disagree, we identify top pixel locations with the highest disagreement scores $a_{\text{DH}}$ and $a_{\text{DM}}$. Features from these regions $fm_{\text{noise}}$ potentially contain both real cells and noise, as represented in Eq. 2 (annotated as $\phi_2$ in Fig. 2).

$$
\begin{aligned}
a_{\text{DM}} &= c \cdot (1 - y_l) \\
\text{Ind}_{\text{DM}} &= \text{argsort}(-a_{\text{DM}})[: k/2] \\
a_{\text{DH}} &= (1 - c) \cdot y_l \\
\text{Ind}_{\text{DH}} &= \text{argsort}(-a_{\text{DH}})[: k/2]
\end{aligned}
\tag{2}
$$

When aggregating potential noise features $fm_{\text{noise}}$ into the distilled noise feature $f_{\text{noise}}$, we calculate the similarity $s_{\text{cell}}$ between the potential noise features $fm_{\text{noise}}$ and the cell feature $f_{\text{cell}}$. Using a weighted sum, we derive the final noise feature $f_{\text{noise}}$, based on the assumption that noise features in these regions are dissimilar to cell features. The process is defined in Eq. 3 (highlighted as $\phi_3$ in Fig. 2).

$$
\begin{aligned}
fm_{\text{noise}} &= f_D([\text{Ind}_{\text{DM}}, \text{Ind}_{\text{DH}}]) \\
s_{\text{cell}} &= \frac{fm_{\text{noise}} \cdot f_{\text{cell}}}{\|fm_{\text{noise}}\| \|f_{\text{cell}}\|} \\
w &= \text{softmax}(1 - \text{norm}(s_{\text{cell}})) \\
f_{\text{noise}} &= w \cdot fm_{\text{noise}}
\end{aligned}
\tag{3}
$$

We compute the similarity between the feature map $f_D$ and the top cell and noise feature $f_{\text{cell}}$ and $f_{\text{noise}}$, obtaining similarity maps $sim_{\text{cell}}$ and $sim_{\text{noise}}$. The computation are provided in Eq. 4 (labeled as $\phi_4$ and $\phi_5$ in Fig. 2).

$$sim_{\text{cell}} = \frac{f_D \cdot f_{\text{cell}}}{\|f_D\|\|f_{\text{cell}}\|}$$
$$sim_{\text{noise}} = \frac{f_D \cdot f_{\text{noise}}}{\|f_D\|\|f_{\text{noise}}\|} \tag{4}$$

**Consensus-aware Loss Function:** During training, the model is guided to focus on regions where both the AI model and lay annotators agree (CP and CN) while ignoring regions likely to contain noise. By combining the confidence map $c$ and lay annotations $y_l$, CP and CN regions are highlighted, and $sim_{\text{cell}}$ and $sim_{\text{noise}}$ further refine the focus on cell-like regions within DM and DH areas. The final supervised loss is defined in Eq. 5:

$$\omega_c = \exp(c \cdot y_l + (1 - c) \cdot (1 - y_l)) \quad \omega_{\text{sim}} = \exp(sim_{\text{cell}} - sim_{\text{noise}})$$
$$\mathcal{L}_{\text{supervise}}(y_l, f(x; \theta)) = (\mathcal{L}_{\text{Dice}} + \mathcal{L}_{\text{BCE}})(y_l, f(x; \theta)) \cdot \omega_c \cdot \omega_{\text{sim}} \tag{5}$$

Where $f$ is the segmentation model, $\theta$ are the trainable parameters, and $\mathcal{L}_{\text{Dice}}$ and $\mathcal{L}_{\text{BCE}}$ are the Dice efficiency loss and Binary Cross-Entropy loss, respectively.

### 2.3. Contrastive Noise Separation

Using the final cell feature $f_{\text{cell}}$ and noise feature $f_{\text{noise}}$, we aim to maximize their separation using a contrastive learning loss function in Eq. 6:

$$\mathcal{L}_{\text{contrastive}}(f_{\text{cell}}, f_{\text{noise}}) = (\mathcal{L}_{\text{KL}} + \mathcal{L}_{\text{MSE}})(\text{norm}(f_{\text{cell}}), \text{norm}(f_{\text{noise}})) \tag{6}$$

where $\mathcal{L}_{\text{KL}}$ is the KL Divergence loss, and $\mathcal{L}_{\text{MSE}}$ is the Mean Squared Error loss.

The final consensus-aware self-corrective learning loss combines $\mathcal{L}_{\text{supervise}}$ and $\mathcal{L}_{\text{contrastive}}$ to achieve robust training shown in Eq. 7:

$$\mathcal{L}_{\text{consensus-aware}}(y_l, f(x; \theta)) = \mathcal{L}_{\text{supervise}}(y_l, f(x; \theta)) + \mathcal{L}_{\text{contrastive}}(f_{\text{cell}}, f_{\text{noise}}) \tag{7}$$

## 3. Data and Experiment

### 3.1. Data

To evaluate the performance of the consensus-aware self-corrective learning framework, we collected a glomerular cell segmentation dataset. We utilized 21 whole slide images (WSIs) from normal adult cases in the nephrectomy dataset and HuBMAP. These slides were stained with Periodic Acid-Schiff (PAS), and were scanned at 20× magnification. The WSIs were cropped into $512 \times 512$-pixel segments to facilitate cell labeling. The cell labels are confined within glomeruli. The labeled cells included mesangial cells (Mes.), endothelial cells (Endo.), podocytes (Pod.), and parietal epithelial cells (Pecs.). Labeling was performed in a partial-label manner, where each image contained a single class label with binary masks. The details of data collection are shown in Table 3 (In Appendix A).

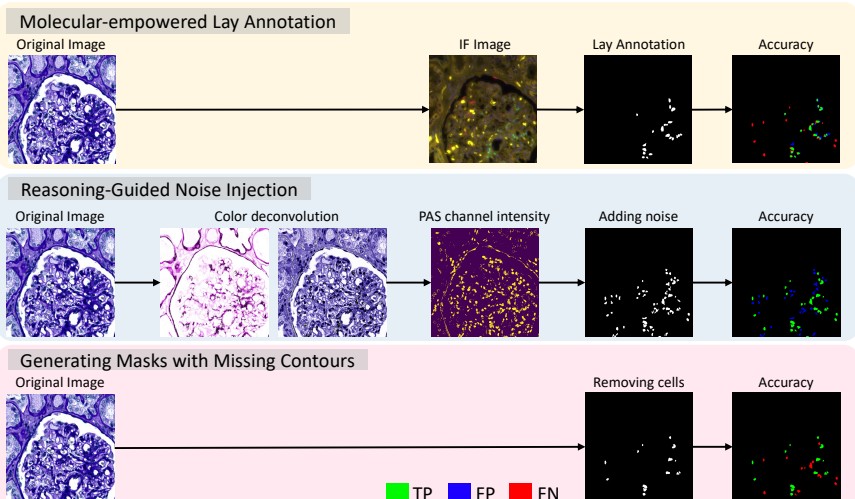

Figure 3: **Illustration of the Noisy Dataset.** The figure depicts a real lay annotation dataset and two reasonable noise generation pipelines used to create FP and FN datasets with plausible noise. These processes are applied to evaluate the proposed method under challenging scenarios.

**Real Lay Annotation Dataset:** Following the annotation process described in (Deng et al., 2023), two sets of annotations were obtained (1) directly from lay annotators and (2) underwent a quality assurance process conducted by experienced pathologists.

**Reasoning-Generated Noise Datasets:** To further explore the capabilities of the proposed method, we designed two reasoning-based noise generation pipelines to create FP and FN datasets: (1) The **FP data generation pipeline** adds plausible noise labels by following these principles: a. annotating nuclei regions indicated by PAS staining; b. providing annotations for glomeruli that are near to the correct cells; and c. creating annotations with sizes that do not exceed the acceptable range for cells, where such annotations are more likely to contain human errors; (2) **The FN data generation pipeline** randomly removes parts of the ground truth labels annotated by pathologists.

The visualizations of the three datasets are shown in Fig. 3, and the labeling accuracy for each dataset, the detailed pipelines are presented in Table 4, Algorithm 1, and Algorithm 2 in the Appendix A.

## 3.2. Experimental details

The dataset was split into training, validation, and testing sets at the WSI level in a 6:1:3 ratio, ensuring balanced distributions of injured and normal glomeruli across splits. All experiments used the same hyperparameter settings, which were determined from an ablation study (see Table 5) on a non-error dataset using supervised learning. Model selection was based on the mean Dice score across the four cell classes in the validation set. All experiments were conducted on an NVIDIA RTX A6000 GPU for uniformity.

### 3.3. Evaluation Metrics

We evaluate performance using Dice similarity coefficient scores, with the binary mask for each image serving as the ground truth. We also provide F1-score results by converting the binary segmentation labels into instance segmentation labels following the method in (Deng et al., 2025). Standard deviations are provided for the results in the tables, and a Wilcoxon t-test is performed to assess the significance of differences between methods.

## 4. Results

### 4.1. Testing Set Segmentation Performance

We evaluate the proposed CASC-AI framework alongside other loss correction noisy label learning methods on three datasets. All methods were implemented with the same backbone and hyperparameters to ensure fair comparisons. We conducted an ablation study to identify the optimal backbone and hyperparameter settings for cell segmentation, using error-free ground-truth labels that were corrected and verified by pathologists under supervised learning, shown in the Appendix B.

Table 1 and Fig. 4 demonstrate that the proposed method achieves improvements compared to direct supervised learning and other baseline methods. This indicates that CASC-AI effectively leverages lay annotations while mitigating noise for enhanced segmentation performance.

Table 1: Performance of various noisy label learning methods. Dice similarity coefficient scores (%) and F1-scores (%) are reported. The top two performing methods are highlighted in red and blue. The Wilcoxon signed-rank test was performed using CASI-AI as the reference method to compare with other methods. All results are statistically significant ($p < 0.001$) compared to the proposed method.

| Real Dataset | | | | | | | | | | |
|---|---|---|---|---|---|---|---|---|---|---|
| **Method** | Dice (%) | | | | | F1-score (%) | | | | |
| | Pod. | Mes. | Endo. | Pecs. | Mean | Pod. | Mes. | Endo. | Pecs. | Mean |
| Supervised | $71.18 \pm 10.08$ | $68.33 \pm 06.87$ | $51.99 \pm 02.99$ | $76.09 \pm 10.60$ | 66.90 | $43.79 \pm 22.19$ | $42.42 \pm 16.78$ | $04.66 \pm 06.61$ | $53.34 \pm 24.87$ | 36.06 |
| GCE (Zhang and Sabuncu, 2018) | $66.94 \pm 07.97$ | $51.25 \pm 02.47$ | $49.71 \pm 00.27$ | $55.56 \pm 06.83$ | 55.86 | $30.96 \pm 18.98$ | $02.53 \pm 06.86$ | $00.00 \pm 00.00$ | $09.67 \pm 17.67$ | 10.79 |
| NCE+NMAE (Ma et al., 2020) | $49.92 \pm 00.05$ | $49.86 \pm 00.12$ | $49.71 \pm 00.27$ | $49.91 \pm 00.07$ | 49.85 | $00.00 \pm 00.00$ | $00.00 \pm 00.00$ | $00.00 \pm 00.00$ | $00.00 \pm 00.00$ | 00.00 |
| NRDice (Wang et al., 2020) | $71.00 \pm 08.20$ | $52.75 \pm 04.42$ | $49.72 \pm 00.27$ | $65.62 \pm 11.74$ | 59.77 | $44.35 \pm 18.48$ | $06.29 \pm 11.69$ | $00.00 \pm 00.00$ | $33.97 \pm 27.68$ | 21.15 |
| CL (Deng et al., 2023) | $74.00 \pm 09.18$ | $67.26 \pm 06.37$ | $69.53 \pm 07.91$ | $73.89 \pm 11.07$ | 71.17 | $50.62 \pm 21.58$ | $38.66 \pm 16.50$ | $42.01 \pm 18.48$ | $49.65 \pm 25.49$ | 45.23 |
| CASC-AI (Ours) | $74.93 \pm 07.38$ | $68.88 \pm 05.32$ | $72.24 \pm 07.29$ | $75.94 \pm 10.71$ | 73.00 | $52.25 \pm 19.90$ | $43.00 \pm 13.93$ | $46.22 \pm 18.83$ | $55.60 \pm 26.32$ | 49.27 |
| **FP Dataset** | | | | | | | | | | |
| **Method** | Dice (%) | | | | | F1-score (%) | | | | |
| | Pod. | Mes. | Endo. | Pecs. | Mean | Pod. | Mes. | Endo. | Pecs. | Mean |
| Supervised | $71.12 \pm 05.45$ | $64.24 \pm 05.35$ | $64.56 \pm 07.12$ | $70.87 \pm 10.62$ | 67.70 | $21.16 \pm 10.67$ | $35.09 \pm 12.60$ | $33.97 \pm 15.68$ | $37.12 \pm 21.30$ | 31.84 |
| GCE (Zhang and Sabuncu, 2018) | $62.71 \pm 08.27$ | $62.27 \pm 04.99$ | $66.16 \pm 07.27$ | $66.96 \pm 10.97$ | 64.52 | $24.01 \pm 17.75$ | $22.68 \pm 11.48$ | $31.69 \pm 15.98$ | $31.68 \pm 25.01$ | 27.52 |
| NCE+NMAE (Ma et al., 2020) | $49.92 \pm 00.05$ | $49.86 \pm 00.12$ | $49.71 \pm 00.27$ | $49.91 \pm 00.07$ | 49.85 | $00.00 \pm 00.00$ | $00.00 \pm 00.00$ | $00.00 \pm 00.00$ | $00.00 \pm 00.00$ | 00.00 |
| RDice (Wang et al., 2020) | $67.66 \pm 07.48$ | $60.10 \pm 06.23$ | $67.97 \pm 07.82$ | $73.16 \pm 11.60$ | 67.22 | $30.98 \pm 15.61$ | $22.33 \pm 13.90$ | $39.24 \pm 18.17$ | $44.78 \pm 25.33$ | 34.34 |
| CL (Deng et al., 2023) | $65.24 \pm 07.38$ | $65.89 \pm 05.32$ | $69.04 \pm 07.29$ | $73.78 \pm 10.71$ | 68.49 | $30.16 \pm 15.89$ | $35.23 \pm 12.34$ | $42.90 \pm 16.87$ | $46.59 \pm 24.08$ | 38.72 |
| CASC-AI (Ours) | $68.49 \pm 06.98$ | $66.24 \pm 05.55$ | $70.64 \pm 07.38$ | $74.75 \pm 10.41$ | 70.03 | $33.23 \pm 14.57$ | $35.35 \pm 12.47$ | $43.00 \pm 16.87$ | $48.34 \pm 25.35$ | 39.98 |
| **FN Dataset** | | | | | | | | | | |
| **Method** | Dice (%) | | | | | F1-score (%) | | | | |
| | Pod. | Mes. | Endo. | Pecs. | Mean | Pod. | Mes. | Endo. | Pecs. | Mean |
| Supervised | $61.51 \pm 08.26$ | $66.60 \pm 07.52$ | $66.02 \pm 08.85$ | $71.13 \pm 11.12$ | 66.32 | $45.49 \pm 19.26$ | $34.93 \pm 19.05$ | $32.69 \pm 19.64$ | $46.33 \pm 27.06$ | 39.86 |
| GCE (Zhang and Sabuncu, 2018) | $56.17 \pm 05.43$ | $49.86 \pm 00.12$ | $49.71 \pm 00.26$ | $49.91 \pm 00.07$ | 51.41 | $12.47 \pm 15.13$ | $00.00 \pm 00.00$ | $00.00 \pm 00.00$ | $00.00 \pm 00.00$ | 03.12 |
| NCE+NMAE (Ma et al., 2020) | $49.92 \pm 00.05$ | $49.86 \pm 00.12$ | $49.71 \pm 00.27$ | $49.91 \pm 00.07$ | 49.85 | $00.00 \pm 00.00$ | $00.00 \pm 00.00$ | $00.00 \pm 00.00$ | $00.00 \pm 00.00$ | 00.00 |
| NRDice (Wang et al., 2020) | $52.48 \pm 04.48$ | $49.86 \pm 00.12$ | $52.85 \pm 03.81$ | $49.91 \pm 00.07$ | 51.28 | $06.62 \pm 13.02$ | $00.00 \pm 00.00$ | $07.32 \pm 10.78$ | $00.00 \pm 00.00$ | 03.48 |
| CL (Deng et al., 2023) | $71.92 \pm 09.18$ | $67.40 \pm 06.37$ | $70.94 \pm 07.91$ | $73.05 \pm 11.07$ | 70.83 | $46.07 \pm 21.83$ | $39.19 \pm 15.21$ | $45.98 \pm 17.61$ | $50.54 \pm 26.40$ | 45.44 |
| CASC-AI (Ours) | $72.85 \pm 08.47$ | $70.04 \pm 04.98$ | $72.63 \pm 07.78$ | $74.90 \pm 11.07$ | 72.60 | $53.01 \pm 21.36$ | $43.17 \pm 14.98$ | $47.31 \pm 18.60$ | $53.00 \pm 26.16$ | 49.12 |

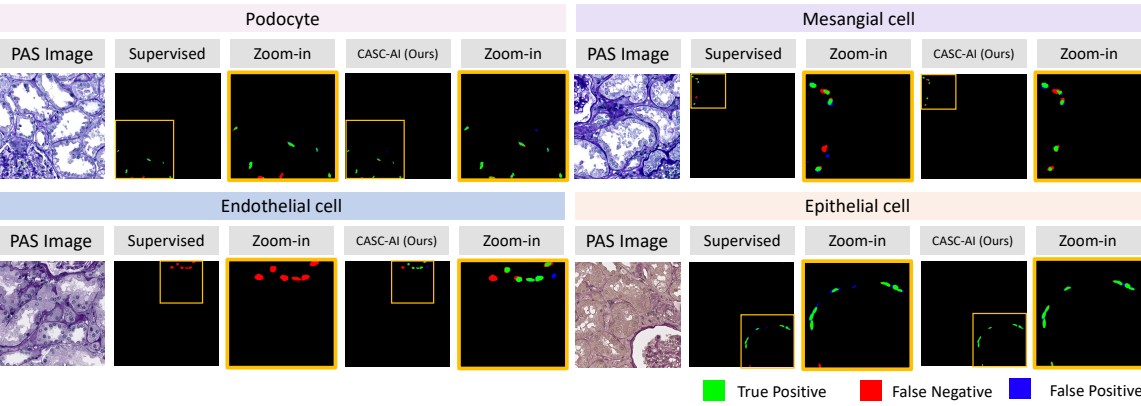

Figure 4: **Qualitative Results.** The figure presents qualitative results on real dataset obtained using the supervised method and the proposed CASC-AI method. The results demonstrate that the proposed approach enhances segmentation performance on noisy labels by reducing false positives and false negatives.

## 4.2. Training Set Segmentation Performance

To evaluate the hypothesis that CASC-AI recognizes FP and FN during training, Table 2 presents Dice scores and F1-score for TP predictions and Intersection over Union (IoU) scores for FP and FN predictions. These results highlight that CASC-AI reduces predictions in FP regions while increasing predictions in FN regions, leading to corrections of the imperfect labels for accurate segmentation during the training phase.

Table 2: Performance on training dataset on TP, FP, and FN regions of the label. Dice similarity coefficient scores (%) and F1-score results (%) are reported on TP, while IoU (%) are reported on FP and FN.

| Method | TP(Dice) ↑ | TP(F1) ↑ | FP(IoU) ↓ | FN(IoU) ↑ | TP(Dice) ↑ | TP(F1) ↑ | FP(IoU) ↓ | TP(Dice) ↑ | TP(F1) ↑ | FN(IoU) ↑ |
|---|---|---|---|---|---|---|---|---|---|---|
| Supervised | 67.99 | 39.25 | **2.86** | 8.20 | 67.35 | 52.63 | 20.67 | 66.52 | 35.58 | 17.01 |
| CASC-AI (Ours) | **73.25** | **48.17** | 3.48 | **9.89** | **69.83** | **56.28** | **18.22** | **68.20** | **38.66** | **18.73** |

## 5. Conclusion

In this work, we present the CASC-AI framework, a consensus-aware self-corrective learning designed to address the challenges of cell segmentation in noisy datasets. By leveraging the Consensus Matrix to identify and prioritize consensus regions between human annotators and the AI model, while adaptively weighting disagreement areas, the framework enhances segmentation reliability even in the presence of noisy annotations. This approach highlights the potential of incorporating an AI model to correct human errors in the labels, paving the way for scalable and robust solutions in medical imaging and digital pathology. The limitations and future work of this study can be found in Appendix C.

## Acknowledgments

This research was supported by the WCM Radiology AIMI Fellowship, NIH R01DK135597 (Huo), DoD HT9425-23-1-0003 (HCY), and the KPMP Glue Grant. This work was also supported by the Vanderbilt Seed Success Grant, the Vanderbilt Discovery Grant, and the VISE Seed Grant.

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

## Appendix A. Data Collection and Experiments

### A.1. Data Information

The details of the patch-level data collection are provided in Table 3.

Table 3: Summary of data collection for different cell classes.

| Class Name | Abbreviation | Patch # | Size | Scale | Stain |
|---|---|---|---|---|---|
| Podocytes | Pod. | 1147 | $512^2$ | $20\times$ | PAS |
| Mesangial cells | Mes. | 789 | $512^2$ | $20\times$ | PAS |
| Glomerular endothelial cells | Endo. | 715 | $512^2$ | $20\times$ | PAS |
| Parietal epithelial cells | Pecs | 2014 | $512^2$ | $20\times$ | PAS |

### A.2. Reasoning-Generated Noise Pipeline

The detailed processes of Reasoning-Guided Noise Injection for FP data and Generating Masks with Missing Contours for FN data are illustrated in Algorithm 1 and Algorithm 2.

### A.3. Label Accuracy

To illustrate the accuracy of the training data, we provide Dice scores and F1-scores for label accuracy in Table 4, compared with noise-free ground truth confirmed by two pathologists.

Table 4: Label accuracy of each dataset. Dice similarity coefficient scores (%) and F1-score results (%) are reported.

| Dataset | Dice (%) | | | | | F1-score (%) | | | | |
|---|---|---|---|---|---|---|---|---|---|---|
| | Pod. | Mes. | Endo. | Pecs. | Mean | Pod. | Mes. | Endo. | Pecs. | Mean |
| Real data | 83.13 | 76.03 | 57.38 | 57.95 | 66.93 | 84.18 | 81.34 | 58.96 | 59.53 | 68.89 |
| FP data | 57.18 | 57.62 | 66.85 | 78.94 | 68.34 | 66.43 | 66.72 | 66.93 | 76.26 | 70.85 |
| FN data | 69.02 | 69.54 | 71.59 | 73.59 | 71.52 | 69.44 | 69.29 | 71.85 | 74.32 | 71.95 |

## Appendix B. Ablation Study

We conducted an ablation study to identify the optimal backbone and hyperparameter settings for cell segmentation, using non-error ground-truth labels that were corrected and verified by pathologists under supervised learning. Results shown in Table 5 indicate that reducing the learning rate to $10^{-4}$ provides the best performance. Increasing the loss weights for the cell class during loss calculations and extending the training epochs did not lead to further performance gains. Our proposed backbone outperformed alternative approaches on our dataset.

## Appendix C. Limitations & Future Work

This study has several limitations. We restricted the design to a **loss-correction approach**. Exploring additional paradigms of corrective learning, such as multi-network architectures or co-training, could further enhance performance. **Exploring additional backbones including instance segmentation models** represents a promising direction for better capturing subtle patterns between cells and noise at the latent level, which could improve overall cell segmentation performance and feature embedding quality. Furthermore, **analyzing noise distributions and patterns, learning the variances among different raters, and incorporating annotator confidence within datasets as conditional information during model training** could provide valuable insights and improve overall noise-label learning, addressing issues such as boundary errors and label ambiguity. Eventually, molecular-empowered cell quantification could be fully automated, from data annotation to AI model training, without any human intervention.

Table 5: Performance on different hyperparameter settings. Dice similarity coefficient scores (%) are reported.

| Backbone | Freeze | Max Epoch | Learning Rate | Loss Weights | Pod. | Mes. | Endo. | Pecs. | Mean |
|---|---|---|---|---|---|---|---|---|---|
| PrPSeg (Deng et al., 2024b) | | 100 | $10^{-3}$ | 1:1 | 73.65 | 68.99 | 70.06 | 73.73 | 71.61 |
| PrPSeg (Deng et al., 2024b) | | 100 | $10^{-3}$ | 10:1 | 73.06 | **71.24** | 70.05 | 72.39 | 71.69 |
| PrPSeg (Deng et al., 2024b) | | 200 | $10^{-3}$ | 1:1 | 74.22 | 70.33 | 69.88 | 74.93 | 72.34 |
| PrPSeg (Deng et al., 2024b) (Ours) | | 100 | $10^{-4}$ | 1:1 | 73.92 | 69.19 | **74.52** | **77.30** | **73.73** |
| PrPSeg (Deng et al., 2024b) | | 200 | $10^{-4}$ | 1:1 | **75.01** | 67.79 | 74.33 | 76.70 | 73.46 |
| PrPSeg (Deng et al., 2024b) | | 100 | $10^{-5}$ | 1:1 | 68.52 | 64.09 | 69.44 | 75.17 | 69.31 |
| CellViT (Hörst et al., 2024) | | 100 | $10^{-4}$ | 1:1 | 57.52 | 51.61 | 49.70 | 58.39 | 54.31 |
| CellViT (Hörst et al., 2024) | Encoder | 100 | $10^{-4}$ | 1:1 | 50.93 | 59.14 | 52.37 | 54.47 | 54.23 |
| CellSAM (Israel et al., 2024) | | 100 | $10^{-4}$ | 1:1 | 49.91 | 49.86 | 49.71 | 49.91 | 49.85 |
| CellSAM (Israel et al., 2024) | Encoder | 100 | $10^{-4}$ | 1:1 | 49.91 | 49.86 | 49.71 | 49.91 | 49.85 |

**Algorithm 1:** Reasoning-Guided Noise Injection (FP data)

---

**Input:** Pathological image $X$, Manual label $Y$, Intensity threshold $T$, Noise limit
       `limit`

**Output:** Processed image and noise mask

Load the pathological image $X$ and corresponding manual label $Y$;

Perform color deconvolution on $X$ to compute stain-specific masks;

Select the PAS channel image and generate a binary mask $M$ using intensity threshold $T$;

Extract contours from $M$ and sort them by proximity to existing annotations in $Y$;

Determine the noise addition limit based on the number of cells in $Y$;

**foreach** *`new_contour` in sorted contours* **do**

    **if** *`new_contour` overlaps with existing annotations or violates spatial constraints*
    **then**

        **continue**;

    **else**

        **if** *`new_contour` size is outside the acceptable range for cells* **then**

            **continue**;

        **else**

            Add `new_contour` to the final noise mask;

        **end**

    **end**

    **if** *number of added contours reaches `limit`* **then**

        **break**;

    **end**

**end**

Save the processed image and the generated noise mask;

---

---

**Algorithm 2:** Generating Masks with Missing Contours (FN data)

---

**Input:** Image $X$, Binary mask $M$, Missing ratio `missing_ratio`
**Output:** Processed image and modified mask
Load the image $X$ and the binary mask $M$;
Extract contours from $M$;
Shuffle the contours randomly;
Set `limit` $\leftarrow (1 - $ `missing_ratio` $) \times$ `len(contours)`;
Initialize `new_mask` $\leftarrow 0$;
Initialize `cnt` $\leftarrow 0$;
**foreach** *contour in contours* **do**
    Draw `contour` on `new_mask`;
    Increment `cnt`;
    **if** *cnt reaches limit* **then**
        **break**;
    **end**
**end**
Save the processed image and the generated noise mask;

---

