# OpenReview forum: "CASC-AI: Consensus-aware Self-corrective Learning for Cell Segmentation with Noisy Labels"
_MIDL.io/2025/Conference — MIDL 2025 Oral_

### Official Review · Reviewer_BSPF · 2025-02-21

**Confidence:** 4
**Preliminary Rating:** 4
**Recommendation:** Poster

**Summary:**

This paper introduces a novel framework for robust multi-class cell segmentation in high-resolution medical images using noisy, lay-annotated data. It utilizes a consensus matrix to prioritize regions of high-confidence segmentation and adaptively weight areas of disagreements (combine with contrastive learning) to identify most reliable features. The framework has been tested on real-world as well as simulated datasets and showed the potential to reduce reliance on pathologist annotations, making this traditionally tedious task of cell segmentation more accessible and scalable.

**Strengths:**

Following are the major strengths for me from this study:
1) The fundamental problem in many AI based digital pathology analysis pipelines is its over reliance on expert annotations. The limited availability of such datasets for each and every cell segmentation task (there is a wide variety of such tasks out there!) makes data driven solutions less accessible; unless approaches like the one proposed are developed and implemented at scale. Such methods truly democratize AI for tasks which were traditionally very isolated.
2) In terms of the techniques, the methodology is novel. The introduction of the consensus matrix combined with contrastive learning to handle noisy annotations is indeed unique. The framework also provides solutions to segmentation noise (robustness of algorithm) using adaptive weight refinement based on the similarity of falsely annotated regions with high-confidence TP annotations.

**Weaknesses:**

To me, the following would be some high level weaknesses to the methodology presented in the manuscript:
1) The reason-guided noise injection takes a very cursory approach to simulating noise in such datasets. There are multiple other ways in the literature that can simulate noise that have not been considered. For example, other major types of noises that could have been included or introduced could be boundary errors or even label ambiguity from lay-annotators in the case of such multi-class segmentation problem.
2) The framework's potential was limited by the limitations of the backbone used in this study. There exists several state-of-the-art transformer based models for cell segmentation that could have been utilized for this task (CellViT, etc.).
3) Another interesting opportunity that the authors had here, was incorporating the idea of ranking raters based on the performance (in terms of confidence) for each task (multi-class labels). Rater's confidence for each task could be estimated using probabilistic models to estimate annotator reliability and class-specific confusion matrices. This can be dynamically weighed during training where the annotator quality scores drive the weights for contribution of each annotation to the loss function.
4) The limited variety of metrics used to assess the model's performance is a weakness. While Dice and IoU are useful metrics, they are not sufficient on their own for evaluating complex segmentation tasks like multi-class cell segmentation. Incorporating additional metrics like Hausdorff Distance, Precision/Recall, Object-Level Metrics, and Panoptic Quality would provide a more comprehensive evaluation of the CASC-AI framework's performance. This would help identify strengths and weaknesses in the model's ability to handle noisy annotations, rare cell types, and boundary accuracy, ultimately leading to a more robust and reliable solution.

**Detailed Comments:**

There are some minor comments that may be immediately addressable that I would also like to point out,
1) Provide some qualitative examples of segmentations from different methods, annotators, etc, for readers to qualitatively gauge the performance of these methods.
2) Discuss the implication of annotator quality. This can be done by probably providing the confidence rating of each rater in terms of the individual segmentation tasks. Comment on how that affects the outcomes. This also points to the 4th weakness discussed above.
3) Discuss the noise generation process in greater detail in your appendix section. This should help the reader understand the realism of the simulated noise and assess whether it accurately reflects real-world annotation errors.

**Justification Of The Preliminary Rating:**

The topic is of great interest to the community, and there is a urgent need for a solution that democratizes the digital histology annotation process, making AI solutions more accessible for all research areas and groups across the spectrum, however, the results are preliminary in nature. More solid ablations studies, along with application to wider problems is still needed.

**Questions To Address In The Rebuttal:**

Address the comments stated above. Provide more rigorous ablation studies where possible to address the comments. Provide more visual examples of methods performances throughout the paper.

**Special Issue:**

No

---

> ### Author Response · Authors · 2025-03-07
> **Official Comments for Reviewer BSPF**
>
> We sincerely appreciate your review and the constructive comments you provided. We have addressed your concerns with a point-by-point response.
>
> ---
>
> **R3Q1:** The reason-guided noise injection offers only a superficial approach to simulating noise, ignoring other established methods (e.g., boundary errors, label ambiguity).
>
> **A1:**
> We agree that boundary errors and label ambiguity are important, especially in tumor segmentation. In this study, however, our focus is on **spatial errors introduced by lay annotators** (i.e., false negatives where real cells are missed and false positives where noise is added) during the molecular-empowered annotation process. To address this focus, we **provide instance-level F1 scores in Table 1, 2, and 4** to highlight performance improvements. We have clarified the specific errors addressed in the paper and will investigate additional error types in the journal extension. Updated clarification will be provided as follows:
>
> > There are several types of annotation errors introduced by humans, including contour-wise boundary errors and instance-wise location errors. In this study, we mainly focus on instance-wise location errors, where false positive and false negative cells are introduced during the molecular-empowered lay annotation process (as shown in Fig. 1).
>
> ---
>
> **R3Q2:** The framework's potential is limited by the choice of backbone.
>
> **A2:**
> We appreciate this comment. In response, we now include results using two state-of-the-art cell segmentation backbones **(CellViT and CellSAM)** under two different training settings. Updated results will be provided as follows:
>
> > We conducted an ablation study to identify the optimal backbone settings for cell segmentation, using error-free ground-truth labels that were corrected and verified by pathologists under supervised learning. Our proposed backbone (Mean Dice: 73.73) outperformed alternative approaches (CellViT: 54.31, CellSAM: 49.85) on our dataset.
>
> ---
>
> **R3Q3:** The authors could rank annotators by their task-specific confidence (using probabilistic models and confusion matrices) to dynamically weight each annotation’s contribution during training.
>
> **A3:**
> We appreciate this insightful suggestion. We agree that incorporating annotator confidence is an interesting direction for future research. We plan to explore this approach in our journal extension and have added a discussion of this concept to the limitations and future work section.
>
> ---
>
> **R3Q4:** The model's evaluation is limited by using only Dice and IoU metrics, which are insufficient for the complexity of multi-class cell segmentation.
>
> **A4:**
> We have now converted these labels to instance segmentation labels and **provide the corresponding F1 score results in Table 1, 2, and 4** following the method in [1]. Additionally, we report **standard deviations and perform a Wilcoxon t-test** to assess the significance of differences between methods.
>
> > The proposed CASC-AI outperformed all baseline methods in every class across three datasets in terms of F1-score, with a p-value of less than 0.001.
> >
> > - Real data — Supervised: 36.06, CL: 45.23, CASC-AI: 49.27
> >  - FP data — Supervised: 31.84, CL: 38.72, CASC-AI: 39.98
> >  - FN data — Supervised: 39.86, CL: 45.44, CASC-AI: 49.12
> >
> > The instance-level results further demonstrate the capability of the proposed method to address false positives and false negatives in the labels.
>
> [1] Deng R, Yao T, Tang Y, et al. KPIs 2024 Challenge: Advancing Glomerular Segmentation from Patch-to Slide-Level. arXiv preprint arXiv:2502.07288, 2025.
>
> ---
>
> **R3Q5:** Provide qualitative examples of segmentation results from different methods and annotators.
>
> **A5:**
> We appreciate the suggestion. We **have added qualitative examples of segmentation results in Figure 4**.
>
> ---
>
> **R3Q6:** Discuss the implications of annotator quality and its effect on outcomes.
>
> **A6:**
> The reviewer raises an important point regarding annotator quality. We acknowledge that incorporating a confidence rating for each annotator could help the model account for individual biases by learning the variance associated with each annotation. We **have added this discussion to the limitations and future work section** and plan to explore this approach in our journal extension.
>
> ---
>
> **R3Q7:** Discuss the noise generation process in greater detail in the appendix to help readers understand the realism of the simulated noise.
>
> **A7:**
> We appreciate this suggestion. We **have expanded the principle of the noise generation process** in the manuscript to provide a more detailed explanation of the reasoning behind the simulated noise, thereby helping readers assess its realism in reflecting real-world annotation errors, which are:
> > Annotations in the glomeruli that are close to the correct cells but not exact.
> > Errors in nuclei regions indicated by PAS staining.
> > Annotations with sizes that do not exceed acceptable limits.

---

### Official Review · Reviewer_C9EL · 2025-02-21

**Confidence:** 4
**Preliminary Rating:** 5
**Recommendation:** Oral
**Final Rating:** 5

**Summary:**

This paper presents a method for training neural networks on noisy labels, specifically for cell segmentation in whole slide images. The noisy labels are obtained from lay annotators, introducing both false positives and false negatives. The proposed Consensus-Aware Self-Corrective AI (CASC-AI) approach weights the loss function higher when the model and annotators agree and distinguishes between likely noise and likely correct annotations in disagreement cases, based on feature similarity to high-confidence agreement regions. The experiments include a synthetic noise dataset, where false positives and false negatives were artificially introduced to better analyze their effects.

**Strengths:**

- The paper addresses a highly relevant problem, obtaining high-quality annotations is challenging, making robust learning methods for noisy data essential.
- The paper is well-written and structured
- The experimental setup is thorough, including a dataset with artificially introduced noise
- The limitations section is detailed

**Weaknesses:**

Minor weaknesses:

- Figure 2 is somewhat dense and takes time to interpret—a simpler overview figure would improve readability.
- In Table 1, abbreviations such as "Pod." and "Mes." should be explained in the caption to improve clarity.
- In Table 4, it would be helpful to highlight the chosen hyperparameters for better visibility.
- Several important details are in the appendix, such as formulas for the consensus-aware self-corrective learning framework. Moving these to the main paper would improve accessibility, while less critical details (e.g., hyperparameter tuning results) could be placed in the appendix instead.

Major weaknesses:
- Hyperparameter tuning details are unclear. Was the search performed in a supervised setting or within the proposed CASC-AI framework? Since all methods share the same backbone and hyperparameters, ensuring a fair comparison by performing hyperparameter search in the supervised setting would be ideal.
- Some Dice scores in the tables are very close, making it difficult to assess significant differences. Providing standard deviations or another measure of variance would improve result interpretation.

**Detailed Comments:**

- Potential journal extension idea: Could the "lay annotator" role also be fulfilled by an AI agent? For example, publicly available foundation models could serve as alternative annotators.
- Would ensembling techniques (e.g., Monte Carlo dropout with mean predictions) achieve similar performance improvements? This could be an interesting additional comparison.

**Justification Of The Final Rating:**

The authors addressed all my comments and improved the manuscript substantially. I appreciate the additional F-1 metric and improved readability by moving the formulas into the main paper. One minor comment: In Figure 1 you could introduce the abbreviations CN,CP, DH, CM in either the middle column or the caption.

**Justification Of The Preliminary Rating:**

This is an interesting and well-written paper on a highly relevant topic, learning from annotations by lay annotators is an important challenge in medical imaging. The proposed method is well-motivated and evaluated, but clarifications on hyperparameter tuning and additional variance measures would improve the paper. Additionally, some key details could be moved from the appendix to the main paper for better understanding.

**Questions To Address In The Rebuttal:**

- clarify which model was used for the hyperparameter tuning
- potentially move the formulas in the appendix to the main paper
- explain abbreviations in Table 1 caption
- add standard deviations to tables

---

> ### Author Response · Authors · 2025-03-07
> **Official Comments for Reviewer C9EL**
>
> We sincerely appreciate your review and the insightful comments you provided. We have addressed your concerns with a point-by-point response.
>
> ---
>
> **R2Q1:** Figure 2 is dense and takes time to interpret; a simpler overview figure would improve readability.
>
> **A1:**
> We appreciate the suggestion. In response, **we have designated Figure 1** with a simpler overview figure to enhance readability.
>
> ---
>
> **R2Q2:** Abbreviations in Table 1 (e.g., “Pod.”, “Mes.”) should be explained in the caption.
>
> **A2:**
> We thank the reviewer for this suggestion. The data description in the manuscript has been updated to **include explanations for these abbreviations**.
>
> > The labeled cells included mesangial cells (Mes.), endothelial cells (Endo.), podocytes (Pod.), and parietal epithelial cells (Pecs.).
>
> ---
>
> **R2Q3:** In Table 4, it would be helpful to highlight the chosen hyperparameters for better visibility.
>
> **A3:**
> We appreciate this suggestion. Table 4 (Now Table 5) has been updated to **clearly highlight the optimal hyperparameter** settings used in our experiments.
>
> ---
>
> **R2Q4:** Several important details (e.g., key formulas) are in the appendix. Moving these to the main paper would improve accessibility.
>
> **A4:**
> We agree with the reviewer’s suggestion and **have moved the key formulas** for the consensus-aware self-corrective learning framework from the appendix to the main manuscript.
>
> ---
>
> **R2Q5:** Hyperparameter tuning details are unclear. Was the search performed in a supervised setting or within the proposed CASC-AI framework?
>
> **A5:**
> All hyperparameter tuning was performed **using supervised learning on a dataset with perfect (noise-free) annotations**. We have clarified this point in the ablation study section.
>
> > We conducted an ablation study to identify the optimal backbone and hyperparameter settings for cell segmentation, using non-error ground-truth labels that were corrected and verified by pathologists under supervised learning.
>
> ---
>
> **R2Q6:** Some Dice scores in the tables are very close; providing standard deviations or another measure of variance would improve result interpretation.
>
> **A6:**
> in response to the reviewers’ concerns, **we now provide F1‐score results** by converting the binary segmentation labels into instance segmentation labels, following the method in [1]. **The full results are shown in Tables 1, 2, and 4.**
>
> Additionally, we **report standard deviations for all results and perform a Wilcoxon t-test** to assess the significance of differences between methods in Table 1. The updated results are as follows:
>
> > The proposed CASC-AI outperformed all baseline methods in every class across three datasets in terms of F1-score, with a p-value of less than 0.001.
> >
> > - Real data — Supervised: 36.06, CL: 45.23, CASC-AI: 49.27
> >  - FP data — Supervised: 31.84, CL: 38.72, CASC-AI: 39.98
> >  - FN data — Supervised: 39.86, CL: 45.44, CASC-AI: 49.12
> >
> > The instance-level results further demonstrate the capability of the proposed method to address false positives and false negatives in the labels.
>
> [1] Deng R, Yao T, Tang Y, et al. KPIs 2024 Challenge: Advancing Glomerular Segmentation from Patch-to Slide-Level. *arXiv preprint arXiv:2502.07288*, 2025.
>
> ---
>
> **R2Q7:** Potential journal extension idea: Could the “lay annotator” role be fulfilled by an AI agent (e.g., publicly available foundation models) as alternative annotators?
>
> **A7:**
> We appreciate this suggestion. We will explore the possibility of using AI-based annotators, such as SAM-based auto-annotation that leverages both molecular and anatomical images, in future work. This discussion has been added to the limitations and future work section.
>
> > Eventually, molecular-empowered cell quantification could be fully automated, from data annotation to AI model training, without any human intervention.
>
> ---
>
> **R2Q8:** Would ensembling techniques (e.g., Monte Carlo dropout with mean predictions) achieve similar performance improvements?
>
> **A8:**
> We agree with the reviewer. We will explore additional ensembling techniques, multi-network approaches, and crowd-sourcing methods in our journal extension. This discussion has been added to the limitations and future work section.
>
> > Furthermore, analyzing noise distributions and patterns, learning the variances among different raters, and incorporating annotator confidence within datasets as conditional information during model training could provide valuable insights and improve overall noise-label learning, addressing issues such as boundary errors and label ambiguity.

---

> ### Author Response · Authors · 2025-03-12
> **Response to Reviewer Final Comments**
>
> Thank you for your positive feedback and for acknowledging the improvements we made to the manuscript. We will update the middle column to introduce the abbreviations in the final version of the manuscript. Thank you once again for your constructive review and valuable suggestions.

---

### Official Review · Reviewer_99AY · 2025-02-21

**Confidence:** 3
**Preliminary Rating:** 2
**Final Rating:** 4

**Summary:**

For cell segmentation, the authors propose to use non-expert annotations, assuming they will be noisy, and then train a segmentation model to deal with the noise by looking into the confusion matrix during training: regions of disagreement between annotations and model's predictions are not necessarily mistakes, since annotations are assumed to be noisy, so they are not penalized as much as they would if annotations were reliable.

**Strengths:**

Being able to learn from crowd-sourced annotations is specially relevant in tasks like this (cell segmentation), where annotation is extremely time consuming and slow. Of course it then becomes meaningful to develop noisy label learning methods in this context.

**Weaknesses:**

- The aspect of this paper that I like the least is that it is filled with weird, non-standard terminology that appears to be unjustified. Why is a segmentation model called here an "AI agent"? AFAIK, AI agents are models that interact with their environment, usually trained with some version of Reinforcement Learning. How is a cell segmentation model, even if trained on noisy labels, an AI agent? I don't think the biomedical computer vision community should buy on this kind of terminology, just because it is everywhere. Please correct me if I misunderstood anything, and there is some agency in the model that I am missing. Other exaggerated terminology includes "Disagreement Overreach" to refer to the model's false positives, or "Reasoning-Generated Noise Datasets" to refer to the artificial introduction of False Positives and Negatives simply by copy-pasting/deleting cell annotations.
- Relevant parts of the work are postponed to the appendix, resulting in weirdly short sections, e.g. see 2.2.1 which has four lines and defers to appendix A to learn the details of two important components of the method. The paper is then full of not-so-interesting ablation studies, like varying the learning rate and seeing the result, or an analysis on performance over the training set, so lack of space is hardly an issue here.
- At times the notation is more python than mathematical, see eq. (1).
- It does not make lots of sense to use a global overlap metric (Dice) to evaluate a cell segmentation model, instance segmentation metrics are much more suitable.

**Detailed Comments:**

Please see above.

**Justification Of The Final Rating:**

Okay, so the authors have invested lots of effort in improving the original submission. Also, I realize the other reviewers were more enthusiastic than me about this paper. I still do not like the attempt the authors make to use overblown terminology; this issue has been partially addressed (most relevantly in the title itself), but not entirely (e.g. what does the reader need to expet from the term "Molecular-empowered Lay Annotation", please?). The paper has improved a lot on the other aspects that me and the other reviewers pointed out (for example the evaluation is more realistic and statistically meaningful), so I am happy to move my rating two steps above until weak accept, congratulations.

Last-minute advice: Table 1 is hardly readable, I would suggest removing the "statistic." columns and just mention in the caption that all results are s.s. with p<0.001, that will make it much cleaner.

**Justification Of The Preliminary Rating:**

The weaknesses stated above are what justify my rating. The language in this paper is confusing and exaggerated, evaluation is (I think) not using appropriate metrics, and the exposition is imbalanced, with relevant technical content postponed to the appendix and not so relevant content taking over lots of space.

**Questions To Address In The Rebuttal:**

- Dice similarity coefficient should be replaced by some other measure of performance that makes more sense in a cell segmentation problem. In the literature (including some work by the authors), versions of the F1 score adapted to this situation are often employed, why not here?
- Are test set annotations made up only of expert-reviewed samples? This is the only way in which performance evaluation would make any sense, if test set labels are also noisy, evaluating on them does not make much sense.
- Why is the language in the paper so unconventional? Am I missing something and this is really a RL-based agent?
- Please use mathematical notation and not pytorch (eq 1)

---

> ### Author Response · Authors · 2025-03-07
> **Official Comments for Reviewer 99AY**
>
> We sincerely appreciate your review and the constructive comments you provided. We have addressed your concerns with a point-by-point response.
>
> ---
>
> **R1Q1:** Why is a cell segmentation model labeled as an “AI agent,” a term usually reserved for models that interact with an environment? Also, please clarify terms like “Disagreement Overreach” and “Reasoning-Generated Noise Datasets.”
>
> **A1:**
> We agree with the reviewers’ comments. Accordingly, we have modified the title to **"Consensus-aware Self-corrective Learning for Cell Segmentation with Noisy Labels"** to better reflect our approach. We have also changed the terms *"Disagreement Overreact"* and *"Missing Consensus"* to *"Disagreement Human-positives"* and *"Disagreement Model-positives,"* respectively, for greater clarity.
>
> For *“Reasoning-Generated Noise Datasets,”* we simulate the most common annotation errors by considering that annotators are most likely to err in the following ways:
>
> > - Providing annotations in the glomeruli that are close to correct cells.
> > - Annotating nuclei regions indicated by PAS staining.
> > - Creating annotations with sizes that do not exceed the acceptable range for cells.
>
> Detailed descriptions of the noise generation process are provided in Algorithms 1 and 2 for reference.
>
> ---
>
> **R1Q2:** Important details are deferred to the appendix (e.g., section 2.2.1), and the paper includes numerous unremarkable ablation studies (e.g., minor variations in learning rate).
>
> **A2:**
> We appreciate the suggestion. In response, **we have moved the key formulas from the appendix into the main manuscript** to improve clarity. Additionally, we now include results using two state-of-the-art **cell segmentation backbones (CellViT and CellSAM)** under two different training settings (training from pretrained model weights and fine-tuning the decoders). **The full results are shown in Table 5**. The updated results are as follows:
>
> > We conducted an ablation study to identify the optimal backbone settings for cell segmentation, using error-free ground-truth labels that were corrected and verified by two pathologists under supervised learning. Our proposed backbone (Mean Dice: 73.73) outperformed alternative approaches (CellViT: 54.31, CellSAM: 49.85) on our dataset.
>
> ---
>
> **R1Q3:** The notation is sometimes more “Pythonic” than mathematical (e.g., Eq. (1)).
>
> **A3:**
> We have removed Equation (1) and replaced it with a conventional description of the confidence score in the text. The updated description is as follows:
>
> > … and a confidence map, which represents the foreground channel of prediction logits after applying the channel-wise softmax function.
>
> ---
>
> **R1Q4:** Global overlap metrics like Dice are not ideal for evaluating cell segmentation; instance segmentation metrics (e.g., adapted F1 scores) are more appropriate.
>
> **A4:**
> The manual annotations for our dataset are provided as partially labeled binary masks for each class. Consequently, we originally framed the problem as a partially-labeled segmentation task and computed binary Dice scores for each image, as instance segmentation labels were not available.
>
> However, in response to the reviewers’ concerns, **we now provide F1‐score results** by converting the binary segmentation labels into instance segmentation labels, following the method in [1]. **The full results are shown in Tables 1, 2, and 4.**
>
> Additionally, we **report standard deviations for all results and perform a Wilcoxon t-test** to assess the significance of differences between methods in Table 1. The updated results are as follows:
>
> > The proposed CASC-AI outperformed all baseline methods in every class across three datasets in terms of F1-score, with a p-value of less than 0.001.
> >
> > - Real data — Supervised: 36.06, CL: 45.23, CASC-AI: 49.27
> >  - FP data — Supervised: 31.84, CL: 38.72, CASC-AI: 39.98
> >  - FN data — Supervised: 39.86, CL: 45.44, CASC-AI: 49.12
> >
> > The instance-level results further demonstrate the capability of the proposed method to address false positives and false negatives in the labels.
>
>
> [1] Deng R, Yao T, Tang Y, et al. *KPIs 2024 Challenge: Advancing Glomerular Segmentation from Patch-to Slide-Level*. arXiv preprint arXiv:2502.07288, 2025.
>
> ---
>
> **R1Q5:** Are test set annotations exclusively expert-reviewed?
>
> **A5:**
> Yes. We confirm that all test set annotations were noise-free, reviewed, and confirmed by two expert pathologists. This information has been added to the experimental details section.

---

> ### Author Response · Authors · 2025-03-10
> **Addressing Reviewer Final Suggestions**
>
> Thank you for recognizing our efforts and recommending acceptance. We would like to address your remaining concerns and suggestions:
>
> (1) We apologize for the unclear definition of “Molecular-empowered Lay Annotation”, which was introduced in the previous study [1]. We will include a more detailed explanation and the appropriate reference to clarify the term.
>
> (2) We agree with your suggestion regarding the visualization of Table 1 and will revise it accordingly.
>
> We will incorporate these changes into the final version of the manuscript.
>
> [1] Deng, R., Li, Y., Li, P., Wang, J., Remedios, L. W., Agzamkhodjaev, S., ... & Huo, Y. (2023, October). Democratizing pathological image segmentation with lay annotators via molecular-empowered learning. In International Conference on Medical Image Computing and Computer-Assisted Intervention (pp. 497-507). Cham: Springer Nature Switzerland.

---

### Author Response · Authors · 2025-03-07
**Official Comments for All Reviewers**

We sincerely thank all the reviewers and Area Chairs for their constructive feedback. This rebuttal addresses the common concerns raised by the reviewers (R1: 99AY, R2: C9EL, R3: BSPF). Detailed point-by-point responses are provided below each reviewer’s comment.

---

**Q1:** The Dice score might not be the best evaluation for cell segmentation, and it is difficult to assess significant differences. (R1, R2, R3)

**A1:**
The manual annotations for our dataset are provided as partially labeled binary masks for each class. Consequently, we originally framed the problem as a partially-labeled segmentation task and computed binary Dice scores for each image, as instance segmentation labels were not available.

However, in response to the reviewers’ concerns, **we now provide F1‐score results** by converting the binary segmentation labels into instance segmentation labels, following the method in [1]. **The full results are shown in Tables 1, 2, and 4.**

Additionally, we **report standard deviations for all results and perform a Wilcoxon t-test** to assess the significance of differences between methods in Table 1. The updated results are as follows:

> The proposed CASC-AI outperformed all baseline methods in every class across three datasets in terms of F1-score, with a p-value of less than 0.001.
>
> - Real data — Supervised: 36.06, CL: 45.23, CASC-AI: 49.27
>  - FP data — Supervised: 31.84, CL: 38.72, CASC-AI: 39.98
>  - FN data — Supervised: 39.86, CL: 45.44, CASC-AI: 49.12
>
> The instance-level results further demonstrate the capability of the proposed method to address false positives and false negatives in the labels.

---

**Q2:** Important formulas are listed in the appendix, while less important ablation studies are in the main text. The hyperparameter tuning details and backbone selection are unclear. (R1, R2, R3)

**A2:**
We appreciate the suggestion. In response, **we have moved the key formulas from the appendix into the main manuscript** to improve clarity. Additionally, we now include results using two state-of-the-art **cell segmentation backbones (CellViT and CellSAM)** under two different training settings (training from pretrained model weights and fine-tuning the decoders). **The full results are shown in Table 5**. The updated results are as follows:

> We conducted an ablation study to identify the optimal backbone settings for cell segmentation, using error-free ground-truth labels that were corrected and verified by two pathologists under supervised learning. Our proposed backbone (Mean Dice: 73.73) outperformed alternative approaches (CellViT: 54.31, CellSAM: 49.85) on our dataset.

---

**Q3:** The term “AI agent” may not correctly indicate the method, and other types of noise (e.g., boundary errors or label ambiguity) are not addressed in the paper. (R1, R3)

**A3:**
We agree with the reviewers’ comments. Accordingly, we have modified the title to **"Consensus-aware Self-corrective Learning for Cell Segmentation with Noisy Labels"** to better reflect our approach. We have also changed the terms *"Disagreement Overreact"* and *"Missing Consensus"* to *"Disagreement Human-positives"* and *"Disagreement Model-positives,"* respectively, for greater clarity.

For *“Reasoning-Generated Noise Datasets,”* we simulate the most common annotation errors by considering that annotators are most likely to err in the following ways:

> - Providing annotations in the glomeruli that are close to correct cells.
> - Annotating nuclei regions indicated by PAS staining.
> - Creating annotations with sizes that do not exceed the acceptable range for cells.

Detailed descriptions of the noise generation process are provided in Algorithms 1 and 2 for reference.

We acknowledge that boundary errors and label ambiguity are also common in medical segmentation, especially in tumor segmentation. In this study, however, **our focus is on the spatial errors introduced by lay annotators** (i.e., false negatives where real cells are missed and false positives where noise is added) during the molecular-empowered annotation process. To address this, we now provide instance-level F1 score results in Tables 1, 2, and 4 to highlight performance improvements. We have clarified the specific errors addressed in the paper as follows:

> There are several types of annotation errors introduced by humans, including contour-wise boundary errors and instance-wise location errors. In this study, we mainly focus on instance-wise location errors, where false positive and false negative cells are introduced during the molecular-empowered lay annotation process (as shown in Fig. 1).

[1] Deng R, Yao T, Tang Y, et al. KPIs 2024 Challenge: Advancing Glomerular Segmentation from Patch-to Slide-Level. *arXiv preprint arXiv:2502.07288*, 2025.

---

### Author Rebuttal · Authors · 2025-03-07

**Rebuttal:**

We sincerely thank all the reviewers and Area Chairs for their constructive feedback, which has significantly helped improve the quality of the manuscript. We have modified the manuscript according to the reviews to address all of the concerns and comments proposed by the reviewers. **All modifications are highlighted in orange. All formatting strictly follows the MIDL LaTeX template.**

The main modifications in the paper are:

1. In response to the reviewers’ concerns, we now **provide F1‐score results** by converting the binary segmentation labels into instance segmentation labels. **Qualitative results** are provided. **Standard deviations** for all results and **Wilcoxon t-test results** are reported to assess the significance of differences between methods. More comprehensive **backbone comparisons** are also included in the ablation study.

2. As suggested by the reviewer, we modified the paper title to **"Consensus-aware Self-corrective Learning for Cell Segmentation with Noisy Labels"**. We changed the terms in the consensus matrix to *“Disagreement Human-positive”* and *“Disagreement Model-positive”*.

3. As suggested by the reviewer, we have moved the **key formulas** from the appendix into the main manuscript and provided a detailed illustration of **the definition of label noise**, a simpler **method figure**, and an **interpretation of the reasoning-guided noise generation** to improve clarity.

For the common concerns raised by all reviewers, please refer to **"Official Comments for All Reviewers"**. For the point-to-point responses, please refer to **"Official Comments for Reviewer XXXX"** below each review.

**Supporting Material:**

/attachment/c3ac05b8864bbe530ecc91fddea5efe2b41a9ec1.pdf

---

### Comment · Area_Chair_ascC · 2025-03-08
**author's responses are available now**

Dear Reviewers,

the rebuttals are available now. Could you take a look at the author's response and update your score please?

Thank you!

---

### Meta-Review · Area_Chair_ascC · 2025-03-19

**Recommendation:** Accept (Poster)
**Confidence:** 4

**Metareview:**

The paper introduces CASC-AI, a self-corrective AI framework for cell segmentation with noisy labels. It leverages a Consensus Matrix to refine learning, improving robustness against annotation noise. The method is validated on real and synthetic datasets, demonstrating significant performance gains.

This work addresses enhancing segmentation robustness through consensus-aware learning. The evaluation is thorough, incorporating instance-level F1 scores, statistical significance testing, and comparisons with state-of-the-art models. The authors have effectively addressed reviewer concerns, refining terminology, expanding evaluation metrics, analysing additional backbone models, and improving the clarity of figures, tables, and explanations.

The paper should be accepted as a poster. While it presents a solid advancement, its application scope makes it better suited for a poster than an oral presentation. The revisions significantly strengthen its impact, making it a valuable addition to the conference.